# Integrated Analysis of Germline and Tumor DNA Identifies New Candidate Genes Involved in Familial Colorectal Cancer

**DOI:** 10.3390/cancers11030362

**Published:** 2019-03-13

**Authors:** Marcos Díaz-Gay, Sebastià Franch-Expósito, Coral Arnau-Collell, Solip Park, Fran Supek, Jenifer Muñoz, Laia Bonjoch, Anna Gratacós-Mulleras, Paula A. Sánchez-Rojas, Clara Esteban-Jurado, Teresa Ocaña, Miriam Cuatrecasas, Maria Vila-Casadesús, Juan José Lozano, Genis Parra, Steve Laurie, Sergi Beltran, Antoni Castells, Luis Bujanda, Joaquín Cubiella, Francesc Balaguer, Sergi Castellví-Bel

**Affiliations:** 1Gastroenterology Department, Institut d’Investigacions Biomèdiques August Pi i Sunyer (IDIBAPS), Centro de Investigación Biomédica en Red de Enfermedades Hepáticas y Digestivas (CIBERehd), Hospital Clínic, 08036 Barcelona, Spain; diaz2@clinic.cat (M.D.-G.); sebasfranch@gmail.com (S.F.-E.); arnau@clinic.cat (C.A.-C.); jenifer.munoz@ciberehd.org (J.M.); bonjoch@clinic.cat (L.B.); annagratacosm@gmail.com (A.G.-M.); paulasanchez10@live.com (P.A.S.-R.); darth.clara@gmail.com (C.E.-J.); mocana@clinic.cat (T.O.); castells@clinic.cat (A.C.); fprunes@clinic.cat (F.B.); 2Systems Biology Program, Centre for Genomic Regulation (CRG), The Barcelona Institute of Science and Technology, 08003 Barcelona, Spain; imagineyd@gmail.com; 3Institut de Recerca Biomedica (IRB Barcelona), The Barcelona Institute of Science and Technology, 08028 Barcelona, Spain; fran.supek@irbbarcelona.org; 4Pathology Department, Hospital Clínic, 08036 Barcelona, Spain; mcuatrec@clinic.cat; 5Bioinformatics Platform, Centro de Investigación Biomédica en Red de Enfermedades Hepáticas y Digestivas (CIBERehd), 08036 Barcelona, Spain; maria.vila.cs@gmail.com (M.V.-C.); juanjo.lozano@ciberehd.org (J.J.L.); 6Centre Nacional d’Anàlisi Genòmica-Centre de Regulació Genòmica (CNAG-CRG), Parc Científic de Barcelona, 08028 Barcelona, Spain; genis.parra@cnag.crg.eu (G.P.); steven.laurie@cnag.crg.eu (S.L.); sergi.beltran@cnag.crg.eu (S.B.); 7Gastroenterology Department, Hospital Donostia-Instituto Biodonostia, Centro de Investigación Biomédica en Red de Enfermedades Hepáticas y Digestivas (CIBERehd), Basque Country University (UPV/EHU), 20014 San Sebastián, Spain; luis.bujandafernandezdepierola@osakidetza.eus; 8Gastroenterology Department, Complexo Hospitalario Universitario de Ourense, Instituto de Investigación Sanitaria Galicia Sur, 32005 Ourense, Spain; joaquin.cubiella.fernandez@sergas.es

**Keywords:** colorectal cancer, whole-exome sequencing, predisposition to disease, germline–tumor analysis, mutational signatures, computational genomics

## Abstract

Colorectal cancer (CRC) shows aggregation in some families but no alterations in the known hereditary CRC genes. We aimed to identify new candidate genes which are potentially involved in germline predisposition to familial CRC. An integrated analysis of germline and tumor whole-exome sequencing data was performed in 18 unrelated CRC families. Deleterious single nucleotide variants (SNV), short insertions and deletions (indels), copy number variants (CNVs) and loss of heterozygosity (LOH) were assessed as candidates for first germline or second somatic hits. Candidate tumor suppressor genes were selected when alterations were detected in both germline and somatic DNA, fulfilling Knudson’s two-hit hypothesis. Somatic mutational profiling and signature analysis were also performed. A series of germline-somatic variant pairs were detected. In all cases, the first hit was presented as a rare SNV/indel, whereas the second hit was either a different SNV (3 genes) or LOH affecting the same gene (141 genes). *BRCA2*, *BLM*, *ERCC2*, *RECQL*, *REV3L* and *RIF1* were among the most promising candidate genes for germline CRC predisposition. The identification of new candidate genes involved in familial CRC could be achieved by our integrated analysis. Further functional studies and replication in additional cohorts are required to confirm the selected candidates.

## 1. Introduction

Colorectal cancer (CRC) is one of the most common and lethal malignant neoplasms worldwide, accounting for 8% of all cancer-related deaths [1]. Developed countries are the most affected, with almost 55% of diagnosed cases, although with better survival rates, since 52% of deaths occur in less-developed regions [2]. The lifetime risk of developing CRC is between 5–6%, but an incidence rise is expected in the coming years, due to higher life expectancy.

Genetic and environmental factors are involved in CRC predisposition. Environmental contributors include alcohol, tobacco and fat intake, among others [3]. Inherited genetic variation reaches 35% of susceptibility according to twin studies [4]. Predisposition can be classified according to population frequency and associated disease risk into high- and low-penetrant variants. High-penetrant variants are rare and have a large effect on the predisposition to the disease. Regarding CRC, well-defined genes such as *APC*, *MUTYH*, the DNA polymerases *POLE* and *POLD1* and the DNA mismatch repair (MMR) family (*MSH2*, *MLH1*, *MSH6* and *PMS2*) are affected by these mutations, causing well-known hereditary syndromes (familial adenomatous polyposis, *MUTYH*-associated polyposis, polymerase proofreading-associated polyposis and Lynch syndrome, respectively) [5]. However, only 5% of CRC cases are explained by this kind of variation due to its low frequency in the population. Low-penetrance genetic variation, mainly identified in genome-wide association studies, is characterized by a high prevalence in the population and a weak deleterious effect. However, collectively, all identified low-penetrance variants contribute significantly to CRC susceptibility, accounting for 5–10% of the heritability to this disease [6].

Familial CRC can be defined as a heterogeneous condition defined by patients with a family history for this neoplasia without alterations in the known hereditary CRC genes. Its etiology is not completely understood yet. The genes responsible are likely to be fairly uncommon but penetrant enough to explain the autosomal dominant patterns of inheritance reported [6]. Recent studies identified potentially implicated genes, including *NTHL1*, *GREM1* and *RNF43*, as the most remarkable [7].

The two-hit hypothesis for cancer development was formulated by Alfred G. Knudson in 1971 [8]. Genes with a loss of function followed by a rapid acceleration of the oncogenic phenotype were named tumor suppressor genes (TSGs). Allelic inactivation can take place as a single nucleotide variant (SNV), a short insertion or deletion (indel), an anomalous methylation or a copy number variant (CNV) [9]. Regarding their distribution, duplications are usually more abundant in healthy individuals than deletions because of their commonly milder phenotypic effect [10]. However, considering Knudson’s hypothesis, a putative second hit would involve a deletion, thus leading to the somatic loss of heterozygosity (LOH).

Commonly used in recent years for the cost-effective discovery of pathogenic SNVs and indels, whole-exome sequencing (WES) has also marked a turning point for CNV and LOH detection. Despite the challenge of the uneven coverage distribution along the genome, WES approaches have emerged as a solid option for germline CNV calling [11], recently obtaining significant results in CRC predisposition [12]. Regarding tumor LOH detection, classic approaches were based on microsatellite markers around the gene of interest. However, ALFRED (allelic loss featuring rare damaging), a novel approach using WES data, has been recently developed to predict putative genes affected by LOH. It is a statistical method capable of inferring LOH status by testing for the allelic imbalance between germline and tumor sequencing data [13].

By means of the development of a combined germline–tumor WES analysis, the purpose of this study was the identification of novel candidate genes involved in germline predisposition to familial CRC. The potential TSG role of the selected candidates was assessed according to Knudson’s two-hit hypothesis.

## 2. Results

### 2.1. Two-Hit Prioritization Strategy Identified New Candidate Genes for CRC Germline Predisposition

WES was performed in 18 unrelated familial CRC patients both in germline and tumor DNA. Prior to data analysis, quality control verifications were carried out. All germline samples yielded good results, with a mean coverage higher than 95× in all cases, resulting in approximately 4 gigabases sequenced per sample. However, two of the tumor samples (FAM22 and H461) showed a significant low value of shared exome regions sequenced (Appendix A) and were finally discarded.

An in-house pipeline was used to identify and filter genetic variants, including SNVs, indels, CNVs and LOHs, in germline and tumor WES data. Those rarest and potentially harmful variants with a function compatible with CRC susceptibility were highlighted. The prioritization strategy selected as candidates those genes affected by two hits according to Knudson’s hypothesis and, therefore, those which are susceptible to have a TSG role.

Regarding germline CNVs, after their integrated calling using two different algorithms and frequency filtering, seven different rare variants were selected (five duplications and two deletions) (Appendix A). However, functions and previously linked phenotypes of the affected genes were not sufficiently relevant to CRC, resulting in their not being further considered as putative germline mutational events. On the other hand, SNVs and indels recorded a total of 494 and 42 germline variants, respectively, after filtering. Thus, only the first hits in the form of SNVs and indels were finally taken under consideration, whereas second hits were selected from the whole spectrum of genetic alterations analyzed (SNV, indels and LOHs).

A total of 143 genes carried a germline–tumor pair of potentially disruptive variants in our samples. Among them, a germline SNV followed by a different somatic SNV was identified in *ADCY8*, *HSPG2* and *TTN*. No indel was found as a tumor second hit. The *TTN* gene encodes for a giant protein of more than 30,000 amino acids, thus having a higher probability of accumulating genetic alterations simply by chance. Considering also its function as a muscular protein, it was discarded as a potential cause for CRC predisposition. Therefore, *ADCY8* and *HSPG2* were selected as the most promising candidates from this double germline–tumor disrupting SNV approach (Table 1).

On the other hand, 141 genes were predicted to be affected by LOH as somatic mutational events with an SNV/indel as a germline first hit (133 SNVs and 8 indels) (Appendix A). Interestingly, LOH was also predicted for *HSPG2* gene, thus presenting both kinds of second hits. Among the 141 germline–tumor pairs of potentially disruptive variants, we pursued an additional prioritization process to better select candidate genes with a plausible implication in CRC predisposition. In this regard, manual curation taking into account protein function compatible with CRC or cancer, as well as previously reported links with susceptibility to CRC or other neoplasms, was considered. A summary of the final 16 functionally prioritized candidates for germline SNV and tumor LOH prediction is shown in Table 2. Interestingly, DNA repair was one of most enriched functions among candidates, with 7 out of 16 genes (43.8%) linked to this cellular mechanism (*BRCA2*, *BLM*, *ERCC2*, *PARP2*, *RECQL*, *REV3L* and *RIF1*). It is also interesting to highlight candidate genes involved in hereditary cancer (*BRCA2*, *BLM*, *ERCC2*, *SMARCA4*) or connected to inherited CRC, such as Cowden syndrome (*SEC23B*) and Peutz–Jeghers syndrome (*STK11IP*). Taking this into account, 10 genes were selected as the best candidates for CRC germline predisposition from the approach of germline SNV/indel and somatic LOH including *BRCA2*, *BLM*, *ERCC2*, *PARP2*, *RECQL*, *REV3L*, *RIF1*, *SEC23B*, *SMARCA4* and *STK11IP*.

All variants located in the 12 final candidate genes were validated by the manual inspection of WES data. In addition, a case-control enrichment analysis for the 12 final candidate genes was performed using a publicly available independent cohort of 1006 patients of familial early-onset CRC (CanVar) and the Exome Aggregation Consortium (ExAC) database. We checked if rare and potentially pathogenic variants in the 12 final candidate genes were also present in this CRC cohort and tested if they were more frequent than in the ExAC control dataset. Potentially pathogenic and rare variants were found in CanVar for all 12 genes assessed. *ADCY8*, *BLM*, *BRCA2*, *ERCC2*, *REV3L*, *RIF1*, *SEC23B*, *SMARCA4* and *STK11IP* were highlighted for harboring a significant enrichment in CRC cases for more than 50% of the potentially disrupting variants (Appendix A).

### 2.2. Somatic Mutational Profiling Detected Hypermutated Tumors Compatible with A Germline Defect Etiology

Different somatic specific features were assessed in order to identify possible links with germline CRC predisposition that could help in the selection of the most suitable candidate genes. Tumor mutational burden analysis presented a large number of mutations per sample, with 5 out of 16 samples showing a hypermutated profile with more than 90 mutations per megabase (Mb), and a median of 58.8 mutations per Mb in the whole cohort (Figure 1). One of the hypermutated samples, H466 (96.9 mutations per Mb), was affected by the putative loss of function of a DNA repair-associated gene according to the two-hit prioritization strategy, *RECQL*. A germline deficiency in the DNA repair pathways affected by this gene could explain both the inherited predisposition to CRC and the elevated tumor mutational prevalence shown by the patient. An ultrahypermutated sample with more than 500 mutations per Mb (sample H470) was also identified. Interestingly, no deleterious mutation in *POLE*, *POLD1* or the MMR genes was found in the germline or somatic profile of this patient.

Regarding mutational signatures, the typical profile of a microsatellite-stable and *POLE*-wild-type CRC is shown by the mutational profile reconstruction using the 30 reference signatures of the COSMIC database (Catalogue of Somatic Mutations in Cancer; https://cancer.sanger.ac.uk/cosmic/signatures). This included a strong predominance of clock-like mutational signature 1, directly associated with the age of onset, along with a low prevalence of signatures related with MMR deficiency (signatures 6, 15, 20 and 26) and *POLE* mutations (signature 10). Specifically, signature 1 has been linked with the spontaneous deamination of 5-methylcytosine at NpCpG trinucleotides leading to T/G mismatches which are not repaired before DNA replication and, therefore, predominantly generate C>T mutations. Interestingly, none of the other signatures currently associated with a particular deficiency in a DNA repair pathway (signatures 2 and 13 with APOBEC activity, signature 3 with double-strand break repair via homologous recombination and signatures 18 and 30 with base excision repair) were detected as a relevant contributor in any of the analyzed tumor samples. Mutational signatures 7 and 11 were the other two signatures with a greater prevalence in our cohort, although they contributed just 6% to the profile reconstruction on average. A link between UV light exposure and signature 7 has been consistently demonstrated, whereas signature 11 has been mostly associated with alkylating chemotherapy treatments. Both etiologies were not apparently relevant for CRC germline predisposition.

## 3. Discussion

An integrated germline–tumor WES analysis was performed in 16 unrelated samples after quality control filtering, resulting in the prioritization of 12 new candidate genes for CRC germline predisposition. A germline SNV and a tumor SNV were identified in *ADCY8* and *HSPG2* genes, whereas a germline SNV/indel and somatic LOH of the wild-type allele was predicted for *BRCA2*, *BLM*, *ERCC2*, *PARP2*, *RECQL*, *REV3L*, *RIF1*, *SEC23B*, *SMARCA4* and *STK11IP*.

*ADCY8* is a membrane-bound enzyme that catalyzes the formation of cyclic AMP (cAMP) from ATP. The cAMP pathway was already found to be associated with cancer, with the overexpression of *ADCY3* increasing oncogenic potential in gastric cancer cells [15] and *ADCY8* acting itself as a risk modifier in glioma [16]. *HSPG2* encodes for the perlecan protein, an essential extracellular matrix component. Its effect on CRC was described using cell lines and tumor xenografts and allografts, where an oncogene role promoting tumor growth and angiogenesis was found [17]. Thus, both genes were not in accordance with the TSG role expected for the genes prioritized by our integrated germline–tumor analysis and were therefore discarded as the putative cause of the inherited predisposition to CRC in the affected families.

A role in DNA repair, along with a previous association with hereditary cancer syndromes, drove the prioritization of the germline SNV/indel and somatic LOH candidates. *RECQL* presented both alterations in a patient with a hypermutated tumor, thus suggesting the hypothesis of a deregulation of a DNA repair mechanism causing a rapid increase in the number of tumor mutations. In this case, the *RECQL* variant (p.Pro74_Trp75delinsGlnCys) was not reported in ExAC and had a potential disruptive effect in the protein structure (Table 2). This gene encodes for a DNA helicase belonging to the RecQ family, responsible for the unwinding of double-stranded DNA and therefore implicated in both DNA replication and repair [18]. Thus, the loss of function of *RECQL* would affect the maintenance of chromosomal stability. In this regard, mutations in this gene have already been linked to breast cancer predisposition [19]. Interestingly, other key members of the same protein family have been associated with well-known recessive cancer predisposition syndromes (*BLM*, Bloom syndrome; *RECQL4*, Rothmund–Thompson syndrome; *WRN*, Werner syndrome) [20]. *BLM* was also found to be mutated in the germline DNA of one patient in our cohort and prioritized by our two-hit integrated analysis. However, although the missense mutation found (p.Pro690Leu) was predicted to be deleterious by different in silico tools and located at the helicase domain of the protein, the tumor showed a low mutation burden. This could indicate either a non-significant effect of the identified variant in *BLM* function, or an association with a distinct carcinogenic mechanism, such us chromosomal instability (linked to a high number of CNVs and aneuploidy instead of SNVs). Interestingly, our study highlights the link between CRC and breast cancer predisposition genes, as well as the relevance of the Fanconi anemia pathway, as also underlined by previous studies [21,22,23].

*BRCA2* and *ERCC2* are also linked with classical cancer predisposition syndromes, hereditary breast and ovarian cancer (HBOC) and xeroderma pigmentosum (XP), respectively [20]. In the case of *BRCA2*, the germline frameshift variant found in family FAM20 (p.Tyr1655fsTer15) was classified as pathogenic in ClinVar for HBOC. A role for this variant in the CRC predisposition was also suggested in a previous study using the same cohort [21] and additionally supported by the presence of additional breast cancer patients in the family (Appendix A). Accordingly, the strength of this association discarded the other prioritized gene in the family, *PARP2*, which was also implicated in DNA repair. In addition, *BRCA2* mutations were found to be significantly enriched in the case-control analysis but not *PARP2* mutations. On the other hand, *ERCC2* encodes for a subunit of the DNA helicase in charge of the nucleotide excision repair (NER) mechanism [24]. Homozygous or compound heterozygous mutations in this gene are known to cause XP, a condition responsible for skin cancer predisposition [25]. In a recent study, its association with breast and ovarian cancer susceptibility was also proposed [26]. Interestingly, a specific mutational signature characterized by a broad distribution of nucleotide changes have recently been associated with somatic mutations in *ERCC2* [27]. However, in the somatic analysis performed for the patient harboring germline mutation in this gene (H458), this signature was not identified. In contrast, a strong predominance of age-related signature 1 was found (84% of somatic mutations explained by this mutational source), along with a small contribution of signature 7 (9%) (Figure 1). UV-derived mutations, commonly responsible for the latter signature, are repaired by NER, potentially altered in this case by the *ERCC2* inactivation and thus explain this specific contribution to the somatic mutational profile observed. The germline mutation detected in our study (p.V230I) is affecting the helicase ATP binding domain of the protein and has not been detected in the ExAC database, thus suggesting its potential disruptive effect. In addition, disruptive variants in this gene were found to be significantly enriched in the case-control analysis performed in familial early-onset CRC patients.

*REV3L* and *RIF1* were also prioritized by our integrated analysis and involved in translesion DNA synthesis and nonhomologous end-joining DNA repair mechanisms, respectively [28,29,30]. Both carried potentially pathogenic germline alterations according to the different evidence assessed (Table 2), whereas the corresponding tumors showed a moderately mutated profile (61 and 30.8 mutations per Mb, respectively). In addition, disruptive variants in both genes were found to be more significantly enriched in cases than controls. *REV3L* was prioritized in family FAM3, where also a double inactivation of *SMARCA4* was predicted by our integrated analysis. The somatic LOH status of both alterations were validated for this specific family using Sanger sequencing in previous studies [21,31]. The results did not confirm an LOH of the wild-type allele in the case of *SMARCA4*, whereas it was detected for *REV3L*, thus supporting this gene as a better candidate.

An ultrahypermutated tumor was also found in one patient of our cohort (H470). The high number of somatic mutations detected cannot be explained by classic somatic hypermutation drivers (*POLE*, *POLD1* and the MMR genes) [32], thus suggesting a specific alteration of another DNA repair mechanism responsible for the phenotype. Interestingly, no gene implicated in this cellular mechanism was identified by our integrated analysis. In contrast, *SEC23B* and *STK11IP* were the genes prioritized through our approach for this patient. The specific functions of proteins encoded by these genes are not directly related with CRC, although both are associated to cancer predisposition syndromes. *SEC23B* is implicated in endoplasmic reticulum to Golgi apparatus transport [33], and has also been recently associated with Cowden syndrome [34]. This inherited condition is linked to hamartomatous polyps and elevated susceptibility to different epithelial cancers, being caused by germline mutations in *PTEN* in most cases [34,35]. On the other hand, Peutz–Jeghers syndrome is an autosomal dominant CRC predisposition syndrome also related to hamartomas and is mainly caused by germline mutations in the TSG *STK11* [5]. *STK11IP*, whose function is not currently broadly described, is known to be interacting with *STK11*, and therefore potentially implicated in CRC predisposition [36].

Our development of a germline–tumor prioritization strategy was in accordance with recent recommendations from the Germline/Somatic Variant Subcommittee (GSVS) of the Clinical Genome Resource (ClinGen), on the use of tumor sequencing data for germline variant interpretation [37]. Even if the loss of heterozygosity and second mutation of the alternative allele assessment were not directly recommended for clinical routine, both pieces of evidence supporting the Knudson’s two-hit hypothesis could add a great value in the variant prioritization process in a comprehensive germline–tumor WES study. In fact, the power of this approach have been proven by previous studies using a similar methodology based in two-hit hypothesis assessment [13,38,39,40,41]. In addition, both tumor phenotypic features analyzed, tumor mutational burden and signatures, were recommended to improve the support of the pathogenicity of germline variants by this and additional studies [37,42]. However, no methylation data may impact the assessment of the two-hit hypothesis, missing those genes affected by epigenetic silencing. In any case, further functional studies and replication in additional cohorts will be needed in order to further confirm the identified potential candidates for CRC germline predisposition.

## 4. Materials and Methods

### 4.1. Patients

Eighteen unrelated Spanish patients (one per family) with unaffiliated strong CRC aggregation compatible with an autosomal dominant pattern of inheritance and available germline and tumor DNA samples were selected from a previously described cohort of 71 individuals from 38 families (Appendix A). Families were selected based on the following criteria: three or more relatives with CRC, two or more consecutive affected generations and at least one CRC diagnosed before the age of 60. The entire cohort had germline WES data available from previous studies [12,21,31]. The presence of germline alterations in well-known genes related with hereditary CRC syndromes (*APC*, *MUTYH* and the DNA MMR genes) were previously discarded for all probands. The present study was approved by the Institutional Ethics Committee (register number 2011/6440, date of approval 22/03/2011). Written informed consent was obtained in all cases.

Matched tumor DNA samples were used to perform WES when available with an optimal quantity and quality from our cohort of 38 CRC families. Tumor DNA was isolated from formalin-fixed paraffin-embedded tissue using the QIAamp Tissue Kit (Qiagen, Redwood City, CA, USA) following the manufacturer’s instructions and reaching a percentage of tumor cells of 70–80% among all 18 available samples. Germline DNA samples of other members of the family diagnosed with CRC, advanced adenoma (i.e., lesion size ≥ 1 cm, villous architecture or high-grade dysplasia) or other extracolonic cancers were also used in previous studies for germline segregation.

### 4.2. Whole Exome Sequencing

Germline WES data were available from previous studies [12,21,31]. WES was performed in tumor samples of selected patients using the HiSeq2000 platform (Illumina, San Diego, CA, USA) and SureSelectXT Human All Exon v5 kit (Agilent, Santa Clara, CA, USA) for exon enrichment. Indexed libraries were pooled and massively parallel-sequenced using a paired-end 2 × 75 bp read length protocol.

The quality control of sequencing data was made in all samples previous to their analysis using the Real-Time Analysis software sequence pipeline (Illumina). Additionally, the proportion of all shared exome regions sequenced with a coverage ≥ 10× was evaluated for tumor samples. A good ratio of shared regions with high coverage (≥ 70%) was expected in good-quality samples, whereas low-quality ones were characterized by a significant drop in this percentage.

WES data analysis was performed in accordance with the workflow displayed in Figure 2. The Burrows–Wheeler Aligner (BWA-MEM algorithm) was used for read mapping to the human reference genome (build hs37d5, based on NCBI GRCh37) [43]. PCR duplicates were discarded using the MarkDuplicates tool from Picard, and then indel realignment and base quality score recalibration were performed with the Genome Analysis Toolkit (GATK, Broad Institute, Cambridge, USA) [44].

### 4.3. Variant Calling and Filtering

#### 4.3.1. SNVs and Indels

The GATK tools HaplotypeCaller and MuTect2 were used for SNV and short indels calling for germline and tumor samples, respectively [44]. To improve germline variant filtering with MuTect2, a panel of 71 available germline CRC samples from the whole cohort was used in the case of five of the tumor samples, whereas an in-house pipeline from the CNAG-CRG (Centre Nacional d’Anàlisi Genòmica-Centre de Regulació Genòmica, Barcelona, Spain) was implemented for the rest. Regarding variant annotation, different databases were considered, including SnpEff, ANNOVAR and dbNSFP for pathogenicity and variant position annotation. PhyloP (phyloP46way_placental score ≥1.6), SIFT (prediction of damaging), PolyPhen2 (HumVar prediction of probably damaging or possibly damaging), MutationTaster (prediction of disease-causing or disease-causing-automatic), LRT (prediction of deleterious) and CADD (Phred score ≥15) were used for the pathogenicity prediction of missense variants. Germline WES data was analyzed through an in-house R language pipeline described in previous studies [12,21,31]. Functions related with CRC or cancer in general were prioritized. DNA repair, apoptosis, autophagy, cell growth, cell proliferation, inflammatory response, cell cycle, angiogenesis, cell differentiation, cell adhesion and chromatin modification, among others, were included. Concerning tumor SNVs and indels, a similar filtering pipeline was used, restraining selected variants to those having a coverage ≥10× both in germline and somatic samples, an alternative allelic frequency in the tumor ≥20%, and also selecting truncating or missense variants fulfilling at least three of the missense pathogenicity tools criteria.

#### 4.3.2. Copy Number Variants and Loss of Heterozygosity

The DNAcopy R package was used for the implementation of the circular binary segmentation algorithm [45]. This was required for the fragmentation of the WES data in order to identify genomic regions with an abnormal value of copy number. After segmentation, CoNIFER and Exome Depth were used in germline data for CNV identification as previously described [12], whereas ALFRED was used to predict the LOH of the wild-type allele in tumor samples [13].

### 4.4. Variant Prioritization and Validation

After the automatic filtering process was performed for all variant types considered, a large number of potentially pathogenic alterations were identified for every sample. Thus, an additional prioritization process was required in order to select those actually relevant for the phenotype under study. Taking advantage of the access to both germline and somatic WES data, an integrated strategy based on Knudson’s two-hit hypothesis was developed in order to look for potential TSGs associated with CRC germline predisposition. Genes with a deleterious germline variant (first hit, SNV/indel or CNV) and a second mutational event in the tumor (second hit, SNV/indel or LOH) were thus prioritized.

The prioritization process was completed with an additional stringent functional selection of the candidate genes compatible with the TSG model expected. The most interesting final candidates were manually curated according to functional evidence. In addition, the amino acidic position of the variants within specific functional protein domains was checked using UniProtKB (http://www.uniprot.org/) and InterPro (http://www.ebi.ac.uk/interpro/), as well as a possible 3D protein structure destabilization effect by using the DAMpred tool (disease-associated mutation prediction; https://zhanglab.ccmb.med.umich.edu/DAMpred/). Special attention was paid to genes previously involved in predisposition to CRC and other neoplasms by reviewing the data present in OMIM (Online Mendelian Inheritance in Man; http://www.omim.org/) and ClinVar (https://www.ncbi.nlm.nih.gov/clinvar/).

The final prioritized variants were validated by manual inspection of the WES data with the Integrative Genomics Viewer [46]. This high-performance data visualizer permits the exclusion of any possible sequencing artifacts, especially those due to strand bias. This is the case when the genotype information given by the data from the forward strand and the reverse strand is significantly different [47]. The CanVar browser [48], a resource of variant level frequency data from cancer germline sequencing studies containing 1006 familial early onset CRC patients, was also used to search for additional variants in this independent familial CRC cohort. Only rare variants (ExAC allele frequency < 0.1%) and potentially pathogenic (CADD Phred score > 15) were considered. Variant enrichment was calculated by comparing the number of cases in the CanVar cohort with the number of controls in the ExAC repository using a Fisher’s exact test.

### 4.5. Mutational Profiling and Mutational Signature Analysis

Somatic WES data was also specifically analyzed in order to look for particular tumor features supporting a hypothesis for the inherited predisposition to familial CRC in the samples considered. In this regard, both the tumor mutational burden and mutational signatures were taken into account. The MuSiCa (Mutational Signatures in Cancer) web application was used to perform these analyses [49]. The prevalence of somatic mutations was described as the total number of SNVs per Mb accumulated in a specific sample, assuming that an average WES sample accounts for 30 Mb with acceptable sequencing quality values. With respect to mutational signatures, the original computational framework described by Alexandrov and collaborators was considered [50,51]. Original mutational profiles of the analyzed samples were reconstructed by the non-negative least squares algorithm using the 30 reference signatures described in the COSMIC database [52].

## 5. Conclusions

Our integrated germline–tumor analysis based on Knudson’s hypothesis allowed the identification of new potential genes implicated in the inherited predisposition to CRC. *BRCA2*, *BLM*, *ERCC2*, *RECQL*, *REV3L* and *RIF1* were among the most promising candidates, with some of them previously associated with predisposition syndromes to other cancers. DNA repair was found to be enriched among the genes prioritized by our approach, thus highlighting the importance of this cellular mechanism in germline predisposition to colorectal carcinogenesis.

## Figures and Tables

**Figure 1 cancers-11-00362-f001:**
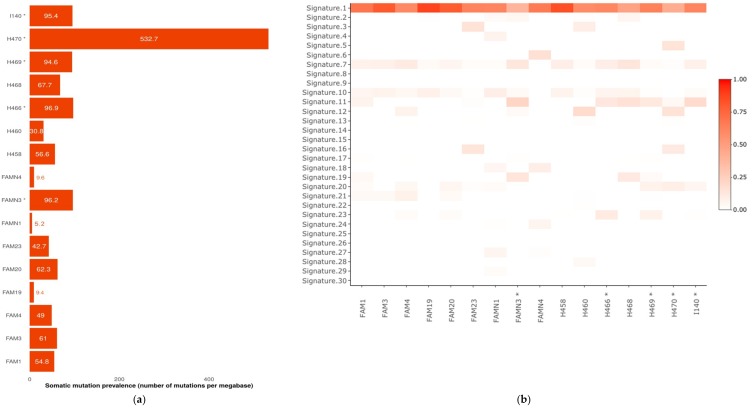
Somatic mutational profile analysis performed with the Mutational Signatures in Cancer (MuSiCa) tool in 16 germline–tumor paired samples. (**a**) Mutational prevalence (number of mutations per sequenced Mb). Hypermutated samples (≥90 mutations/Mb) are marked with an asterisk (*); (**b**) mutational signature refitting analysis showing the contributions of the 30 Catalogue of Somatic Mutations in Cancer (COSMIC) reference mutational signatures in the mutational catalogues of the samples of the study.

**Figure 2 cancers-11-00362-f002:**
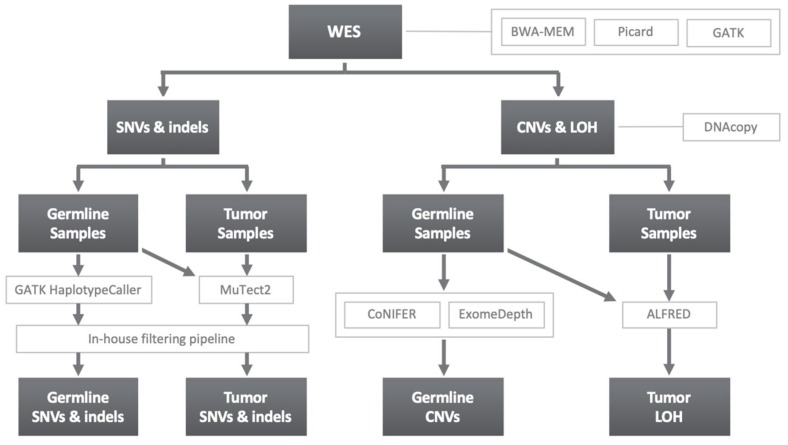
Methodology schematic for variant identification, showing the software used in each analysis step for the different classes of genetic variation considered. WES, whole-exome sequencing; BWA-MEM, Burrows-Wheeler Alignment Tool; GATK, Genome Analysis Toolkit; SNV, single nucleotide variant; indels, insertion and deletion variants; CNV, copy number variant; LOH, loss of heterozygosity; MuTect2, somatic SNV and indel variants caller.

**Table 1 cancers-11-00362-t001:** Description of the genes carrying a potentially disruptive germline SNV (single nucleotide variant) and a different SNV in the matched-tumor sample.

Gene	Family	RefSeq Transcript	Hit	Genetic Variant	Path. Tools	DAMpred	ExAC Freq.	Protein Domain	Protein Function
*ADCY8*	FAMN4	NM_001115.2	1st	c.1747G>A p.(Glu583Lys)	5/6	−	21/60,697	Adenylyl cyclase class-3/4/guanylyl cyclase domain	Biosynthesis of cAMP from ATP
			2nd	c.458C>T p.(Ile153Thr)	4/6	+	0/60,706	Interaction with *ORAI1*, *STIM1*, *PPP2CA* and *PPP2R1A*	
*HSPG2*	FAM23	NM_005529.7	1st	c.3148G>A p.(Gly1050Ser)	3/6	−	3/60,456	Laminin IV type A domain	Component of vascular extracellular matrix, regulation of angiogenesis and cell growth
			2nd	c.7406C>T p.(Thr2469Met)	4/6	−	0/60,706	Immunoglobulin-like C2-type domain	

Abbreviations: DAMpred, disease-associated mutation prediction, affects protein structure (+), no effect on protein structure (−); ExAC, Exome Aggregation Consortium; Freq., frequency; Path., pathogenicity, cAMP: cyclic AMP.

**Table 2 cancers-11-00362-t002:** Candidate genes for germline colorectal cancer (CRC) predisposition selected after the two-hit prioritization strategy. In all cases, a first single nucleotide variant (SNV)/indel hit was present in the germline and a second loss of heterozygosity (LOH) hit was identified in the matched-tumor sample.

Gene	Family	RefSeq Transcript	Genetic Variant	Path. Tools	DAMpred	ExAC Freq.	Protein Domain	Protein Function
*BRCA2*	FAM20	NM_000059.3	c.4963delT p.(Tyr1655fs*15)	FS	n.a.	0/60,706	-	Double-strand break repair via homologous recombination, inherited predisposition to breast and ovarian cancer
*BLM*	FAMN4	NM_000057.4	c.2069C>T p.(Pro690Leu)	6/6	+	1/60,570	Helicase ATP-binding domain	DNA helicase, double-strand break repair via homologous recombination, regulation of cell cycle and apoptosis, DNA replication, telomere maintenance
*ERCC2*	H458	NM_000400.3	c.688G>A p.(Val230Ile)	4/6	−	0/60,706	Helicase ATP-binding domain	DNA helicase, transcription-coupled nucleotide excision repair, regulation of cell cycle
*FAT2*	FAMN3	NM_001447.2	c.1643T>C p.(Val548Ala)	5/6	−	0/60,706	Cadherin domain	Regulation of cell proliferation, cell adhesion
*IGF2R*	H466	NM_000876.3	c.232G>A p.(Gly78Arg)	6/6	+	1/60,684	-	Positive regulation of apoptosis
*LATS2*	H460	NM_014572.3	c.337G>A p.(Asp113Asn)	5/6	−	1/56,138	Ubiquitin-associated domain	Positive regulation of apoptosis, regulation of cell cycle
*PARP2*	FAM20	NM_005484.3	c.910G>C p.(Glu304Gln)	3/6	−	3/60,208	Poly(ADP-ribose) polymerase (PARP) alpha-helical domain	Base excision repair, extrinsic apoptotic signaling pathway
*PSMD9*	H469	NM_002813.6	c.361A>T p.(Ser121Cys)	3/6	−	30/60,148	PDZ domain	Subunit of 26S proteasome, regulation of apoptosis and cell cycle, regulation of ubiquitin-protein ligase activity
*RASSF6*	H460	NM_201431.2	c.779C>T p.(Pro260Leu)	6/6	−	53/60,475	Ras-associating domain	Positive regulation of apoptosis
*RECQL*	H466	NM_002907.4	c.221_225delinsAATGT p.(Pro74_Trp75delinsGlnCys)	6/6	+	0/60,706	-	DNA helicase, double-strand break repair via homologous recombination, DNA replication
*RERGL*	H466	NM_024730.3	c.362T>C p.(Val121Ala)	6/6	+	54/60,446	-	Unknown (closely related to *RERG*, which functions as a negative regulator of cell growth [14])
*REV3L*	FAM3	NM_002912.4	c.559A>T p.(Arg187Trp)	5/6	−	0/60,706	Exonuclease domain (family B of DNA polymerases)	DNA repair, translesion DNA synthesis
*RIF1*	H460	NM_018151.4	c.4262G>A p.(Arg1421His)	4/6	+	5/59,938	-	Double-strand break repair via nonhomologous end joining, telomere maintenance
*SEC23B*	H470	NM_032985.5	c.531G>C p.(Glu177Asp)	4/6	−	1/60,706	Sec23/Sec24 trunk domain	Intracellular protein transport, associated with inherited cancer predisposition Cowden Syndrome
*SMARCA4*	FAM3	NM_003072.3	c.295C>T p.(Arg99Trp)	5/6	−	1/60,196	-	Regulation of cell growth, regulation of cell cycle, chromatin remodeling
*STK11IP*	H470	NM_052902.4	c.1214C>T p.(Pro405Leu)	5/6	−	51/59,930	-	Interaction with *STK11* (serine/threonine kinase activity, negative regulation of cell growth, Peutz-Jeghers CRC predisposition syndrome)

Abbreviations: DAMpred: disease-associated mutation prediction, affects protein structure (+), no effect on protein structure (−); n.a., not available; ExAC, Exome Aggregation Consortium; Freq., frequency; FS, frameshift; Path., pathogenicity.

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
