# Peer review of "Integrated Analysis of Germline and Tumor DNA Identifies New Candidate Genes Involved in Familial Colorectal Cancer"

_cancers, 2019, doi:10.3390/cancers11030362_

Round 1

Reviewer 1 Report

The authors are to be congratulated on a valuable study.

Author Response

Response to Reviewer 1 Comments Point 1: The authors are to be congratulated on a valuable study. Response 1: We thank the reviewer for his/her positive comments about our article.

Reviewer 2 Report

Cancers

Integrated analysis of germline and tumor DNA identifies new candidate genes involved in familial colorectal cancer

Authors: Marcos Díaz-Gay et al

The study analyses genetic variants, including SNVs, indels, CNVs and LOHs in 18 unrelated CRC families. This is an important study which can be used as a baseline for similar analysis in other cancers. Biological validation or atleast verification would have made this a very high impact study.

PROS

This study involved somatic mutational profiling and signature analysis and detected germline somatic variant pairs.  Interestingly BRCA2, BLM, ERCC2, RECQL, REV3L and RIF1 were among the most promising candidate genes for germline CRC predisposition. This integrated analysis to identify new candidate genes involved in familial CRC can be utilized for other cohorts.

The paper is well written, and the parameters are well designed to support the hypothesis and conclusions.

The paper also does well to identify key genes for familial CRC.

Minor

The article is a bioinformatic analysis, functional validation would have been interesting.

The authors fail to cite Cancer Res. 2014 Dec 1;74(23):6947-57, PLoS Genet. 2008 Jul;4(7):e1000115, Stat Appl Genet Mol Biol. 2009;8:Article 47.: which are some established study for the integrated analysis in the field.

Author Response

Response to Reviewer 2 Comments

Point 1: The study analyses genetic variants, including SNVs, indels, CNVs and LOHs in 18 unrelated CRC families. This is an important study which can be used as a baseline for similar analysis in other cancers. Biological validation or at least verification would have made this a very high impact study.

Response 1: We thank the reviewer for this comment. We agree that biological validation of our results would increase the value of our study. However, our plan was first to use this data analysis approach to precisely select which candidate genes should undergo a further functional validation as a next step.

Point 2: This study involved somatic mutational profiling and signature analysis and detected germline somatic variant pairs. Interestingly BRCA2, BLM, ERCC2, RECQL, REV3L and RIF1 were among the most promising candidate genes for germline CRC predisposition. This integrated analysis to identify new candidate genes involved in familial CRC can be utilized for other cohorts. The paper is well written, and the parameters are well designed to support the hypothesis and conclusions. The paper also does well to identify key genes for familial CRC.

Response 2: We thank the reviewer for his/her positive comments about our article and agree on the fact that this kind of data analysis can be tested in our cancer cohorts.

Point 3: The article is a bioinformatic analysis, functional validation would have been interesting. Response 3: We agree again with the reviewer on this comment. As previously mentioned, our plan was first to use this data analysis approach to precisely select which candidate genes should undergo a further functional validation as a next step.

Point 4: The authors fail to cite Cancer Res. 2014 Dec 1;74(23):6947-57, PLoS Genet. 2008 Jul;4(7):e1000115, Stat Appl Genet Mol Biol. 2009;8:Article 47.: which are some established study for the integrated analysis in the field.

Response 4: As requested, we have added the above references in our article.

Reviewer 3 Report

In this paper entitled "Integrated analysis of germline and tumor DNA identifies new candidate genes involved in familial colorectal cancer", the authors analyze 18 samples from as many families with predisposition to CRC, in order to identify possible mutations predisposing the hereditary syndrome. Using WGS and an integrated method of analysis comparing the presence of SNV, Indel and CNV and LOH in the germinal and tumor DNA and selecting those tumor suppressor genes that were altered in both the germinal and somatic DNA (the theory of two Knudson hits).

This is an interesting and well-writtenmanuscript about the study of new genes involved in colorectal cancer predisposition, with a detailed description of the bioinformatics analysis performed on the data.

Comments:

The references about the colorectal cancer incidence (1 and 2) are not the most recent ones. The authors should update them.

In the introduction, the authors go too much into details about the Knudson’s hypothesis and the definition of tumor suppressor genes, SNV, CNV etc. (lines 78-90). They should shorten this part, citing the references and leaving basic definitions to didactic literature. 

For the mutation analyses, only SNV and indel were considered at the germline level, while at the tumor level for the second hit were also considered LOH and CNV. Has no LOH and/or CNV been identified at the germinal level?

In the discussion, the authors highlight the interesting link between colorectal cancer and breast cancer predisposition genes, but they should add some recent papers about the emerging link between BLM, RECQL, and Fanconi anemia genes (e.g. Barroso et al.Breast Cancer Res Treat 2009; Kurian et al.JCO 2014; Tedaldi et al.Oncotarget 2017). 

The authors report the references about the patients’ cohort, that was previously described, but it would probably useful to remind here the geographical origin of the patients and the criteria applied for the patients’ selection.

The pedigree for each family is shown in the article, but the clinical characteristics of the patients are not indicated.

The potential variant pathogenicity was evaluated only by comparison on databases (Exac and CanVar).

Updated HGVS rules is preferred (Three letter aminoacid code, LRG ....)

Author Response

Point 1: In this paper entitled "Integrated analysis of germline and tumor DNA identifies new candidate genes involved in familial colorectal cancer", the authors analyze 18 samples from as many families with predisposition to CRC, in order to identify possible mutations predisposing the hereditary syndrome. Using WES and an integrated method of analysis comparing the presence of SNV, indel and CNV and LOH in the germinal and tumor DNA and selecting those tumor suppressor genes that were altered in both the germinal and somatic DNA (the theory of two Knudson hits). This is an interesting and well-written manuscript about the study of new genes involved in colorectal cancer predisposition, with a detailed description of the bioinformatics analysis performed on the data.

Response 1: We thank the reviewer for his/her positive comments about our article.

Point 2: The references about the colorectal cancer incidence (1 and 2) are not the most recent ones. The authors should update them.

Response 2: We agree with the reviewer and we have updated these references accordingly: 1. Ferlay, J.; Colombet, M.; Soerjomataram, I.; Mathers, C.; Parkin, D.M.; Piñeros, M.; Znaor, A.; Bray, F. Estimating the global cancer incidence and mortality in 2018: GLOBOCAN sources and methods. Int. J. Cancer 2019, 144, 1941-1953. 2. Bray, F.; Ferlay, J.; Soerjomataram, I.; Siegel, R.L.; Torre, L.A.; Jemal, A. Global cancer statistics 2018: GLOBOCAN estimates of incidence and mortality worldwide for 36 cancers in 185 countries. CA Cancer J Clin 2018, 68, 394-424.

Point 3: In the introduction, the authors go too much into details about the Knudson’s hypothesis and the definition of tumor suppressor genes, SNV, CNV etc. (lines 78-90). They should shorten this part, citing the references and leaving basic definitions to didactic literature.

Response 3: As suggested by the reviewer, we have shortened this part. The two-hit hypothesis for cancer development was formulated by Alfred G. Knudson in 1971 [8]. Genes with loss of function followed by a rapid acceleration of the oncogenic phenotype were named tumor suppressor genes (TSGs). Allelic inactivation can take place as a single nucleotide variant (SNV), a short insertion or deletion (indel), an anomalous methylation or a copy number variant (CNV) [9]. Regarding their distribution, duplications are usually more abundant in healthy individuals than deletions because of their commonly milder phenotypic effect [10]. However, considering Knudson’s hypothesis, a putative second hit would involve a deletion, thus leading to the somatic loss of heterozygosity (LOH).

Point 4: For the mutation analyses, only SNV and indel were considered at the germline level, while at the tumor level for the second hit were also considered LOH and CNV. Has no LOH and/or CNV been identified at the germinal level?

Response 4: As stated in the article, seven different germline rare CNV were selected (five duplications and two deletions). They are summarized in Table S1. However, functions and previous linked phenotypes of the affected genes were not sufficiently relevant to colorectal cancer, resulting in not being further considered as putative germline mutational events.

Point 5: In the discussion, the authors highlight the interesting link between colorectal cancer and breast cancer predisposition genes, but they should add some recent papers about the emerging link between BLM, RECQL, and Fanconi anemia genes (e.g. Barroso et al. Breast Cancer Res Treat 2009; Kurian et al.JCO 2014; Tedaldi et al.Oncotarget 2017).

Response 5: As requested, we have added two of the above references in our article (Barroso et al. 2009 and Tebaldi et al. 2017) and the following sentence to the discussion: Interestingly, our study highlights the link between CRC and breast cancer predisposition genes, as well as the relevance of the Fanconi anemia pathway, also underlined by previous studies [21-23].

Point 6: The authors report the references about the patients’ cohort, that was previously described, but it would probably useful to remind here the geographical origin of the patients and the criteria applied for the patients’ selection.

Response 6: As suggested by the referee, we have specified the geographical origin of the patients and the applied criteria for their selection. We have modified the below sentence from the Material and Methods section as follows: Eighteen unrelated Spanish patients (one per family) with unaffiliated strong CRC aggregation compatible with an autosomal dominant pattern of inheritance and available germline and tumor DNA samples were selected from a previously described cohort of 71 individuals from 38 families (Figure S2). Families were selected based on the following criteria: 3 or more relatives with CRC, 2 or more consecutive affected generations and at least one CRC diagnosed before the age of 60. The entire cohort had germline WES data available from previous studies [12,21,31].

Point 7: The pedigree for each family is shown in the article, but the clinical characteristics of the patients are not indicated.

Response 7: We agree with the reviewer on the importance of this issue. However, we believe it is not necessary to profusely describe the clinical characteristics of the patients in this study since they were described in previous articles from our group [12,21,31]. On the other hand, we also considered it was not the main focus of this study, as well as article space was saved and readability was improved by not including again this information.

Point 8: The potential variant pathogenicity was evaluated only by comparison on databases (Exac and CanVar).

Response 8: We also agree with the reviewer on the importance of this issue. As previously mentioned in our replying to Reviewer 2 comments, our plan was first to use this data analysis approach to precisely select which candidate genes should undergo a further functional validation as a next step.

Point 9: Updated HGVS rules is preferred (Three letter aminoacid code, LRG ....).

Response 9: As suggested by the referee, we have modified description of sequence variants accordingly.
